# Vitamin D Status and Associated Factors of Older Adults in the Cross-Sectional 2015–2017 Survey

**DOI:** 10.3390/nu15204476

**Published:** 2023-10-23

**Authors:** Yichun Hu, Rui Wang, Deqian Mao, Jing Chen, Min Li, Weidong Li, Xiaoguang Yang, Lichen Yang

**Affiliations:** 1National Institute for Nutrition and Health, China CDC, Beijing 100050, China; huyc@ninh.chinacdc.cn (Y.H.); wangrui@ninh.chinacdc.cn (R.W.); maodq@ninh.chinacdc.cn (D.M.); chenjing@ninh.chinacdc.cn (J.C.); limin@ninh.chinacdc.cn (M.L.); liwd@ninh.chinacdc.cn (W.L.); yangxg@ninh.chinacdc.cn (X.Y.); 2Key Laboratory of Trace Element Nutrition of National Health Commission, Beijing 100050, China

**Keywords:** vitamin D, Chinese older adults, 25-hydroxyvitamin D, CACDNS 2015–2017

## Abstract

Vitamin D is beneficial for maintaining good health; however, there is a lack of nationally representative data reported, particularly in older adults. To better understand the nutritional status of vitamin D and its influencing factors on Chinese older adults, we adopted stratified random sampling to select serum samples originating from the Chronic Disease and Nutritional Survey Biobank of Chinese Residents in 2015–2017. Serum 25-hydroxyvitamin D (25(OH)D) concentrations were determined by enzyme-linked immunoassay. The OR and PR of associated factors for vitamin D deficiency and insufficiency were calculated. In the study, a total of 6273 participants were included. Median serum 25(OH)D concentration was 18.48 (13.27–24.71) ng/mL. The overall rate of vitamin D deficiency and insufficiency was 58.27% (<20 ng/mL), and the VDD rate was 22.17%, which is worse than 5 years ago by nearly 20%. The likelihood of vitamin D deficiency and insufficiency is increased in women, people aged and above 70 years, ethnic minorities, people living in urban areas, midlands, or western areas, warm or medium temperate zones, with middle school and above education level, and people with abdominal obesity and anemia would increase the possibility of vitamin D deficiency and insufficiency with latitude having the greatest impact on vitamin D deficiency and insufficiency. Overall, vitamin D deficiency and insufficiency are very common in Chinese older adults. They should be encouraged to improve their vitamin D nutritional status through enough sunshine exposure and increasing vitamin D intake through diet or supplements.

## 1. Introduction

Existing research indicates that vitamin D (calcitriol) is an essential fat-soluble vitamin for sustaining health. The traditional role of vitamin D is to support the intestinal absorption of calcium and to maintain appropriate blood calcium and phosphate concentrations required for healthy bone mineralization. Therefore, vitamin D can help prevent osteoporosis along with calcium [1], especially for older adults. More than a billion children and adults worldwide suffer from vitamin D deficiency and insufficiency [2]. In adults, vitamin D deficiency leads to abnormal mineralization of the collagen matrix in bone, known as osteomalacia. Vitamin D deficiency also results in muscle weakness and pain. In addition to its classical role in regulating calcium-phosphorus balance and maintaining bone health, vitamin D has a wider range of extra-osseous biological effects, such as immune regulation, anti-tumor, and protection of the central nervous system and prevention of metabolic syndrome. Vitamin D deficiency has become a global health concern for people of all ages [3,4]. It is reported that vitamin D deficiency (<20 ng/mL) affects 40% to 60% of people with systemic myalgia and bone pain [5].

However, there are few reports on vitamin D nutritional status in nationally representative populations, especially on vitamin D levels in older adults. Since 2010–2012, the Chinese National Nutrition and Health Survey has included vitamin D as a regular monitoring indicator, and our team has been responsible for monitoring the vitamin D status of the whole population in the survey. In our previous reports, we have found a gradual increase in vitamin D deficiency in different populations, such as pregnant women, children, and adolescents; thus, the nutritional status for vitamin D needs to be continuously monitored in all populations.

Older adults experience degenerative changes in their bodies as they age, including diminished cellular function, structural damage, and slowed metabolism. Among different populations, the top risk group for poor adult vitamin D status is older adults. Factors associated with vitamin D levels, such as age, gender, geographic location, and physical condition were analyzed to suggest influencing factors that increase the likelihood of vitamin D deficiency and insufficiency. It is hoped that the findings of the present study will be useful in formulating policy recommendations for vitamin D deficiency interventions.

## 2. Materials and Methods

### 2.1. Subjects and Ethics

The information of participants for this study came from the China Adult Chronic Disease and Nutrition Survey (CACDNS 2015–2017), the sixth round of China’s national nutrition health survey, carried out by the Chinese Center for Disease and Control Center (China CDC) of the non-institutionalized civilian population of China. The survey covered the county (district)-level administrative units (counties, county-level cities, and districts) in all 31 provinces in mainland China. All of the respondents in the CACDNS 2015–2017 were selected using a complex, stratified cluster random selection procedure. Additionally, the bio specimen bank was established. The respondents were selected from the established bio specimen bank by using a random sampling method, and the minimum sample size was calculated and determined based on the reported vitamin D deficiency rate. Respondents aged 60 years old and above with complete basic information were included and those without complete questionnaire information and that had poor blood quality were excluded. Each participant gave their written informed permission before participating in the survey. The National Institute for Nutrition and Health (NINH), China CDC Ethics Review Board approved the protocol (No. 201519, approved on 30 May 2015).

### 2.2. Data Collection and Definition

China CDC set up a national workgroup to carry out the survey. At the site, an interview and anthropometric measures were conducted, and blood were collected. Standardized questionnaires were used to collect the basic data, including age, sex, region type, ethnicity, and marital status. Height and weight were measured by a uniform method and apparatus described by Shi et al. [6]. Calculation of body mass index (BMI) was based on weight (kg) and height (m). Waist circumference (WC) was measured with a soft band at the midpoint of the line between the subcostal arch margin and the iliac crest in the midaxillary region, and the results were averaged over two measurements. Hemoglobin (Hb) concentrations were measured on site at the district (county) CDC laboratories in the survey area. All laboratory operators were required to attend a uniform training and examination. Only after they were qualified could they carry out Hb measurements. All of the data were collected and entered into the systematic platform of the China National Nutrition and Health Survey System developed for the survey. Demographic information (such as age, sex, and race), education, and marital status were recorded from self-reports. All the information was collected and entered into the systematic platform of the CACDNS 2015–2017. The region type was recorded and categorized appropriately [7,8]. With the exception of Han, all ethnicities are considered minorities. The latitude, which is taken from the Baidu map, is separated into tropical (0–23.5 °N), subtropical (23.5–32.0 °N), warm temperate (32.0–40.5 °N), and medium temperate (40.5–46.5 °N) zones. The months of spring (March to May), summer (June to August), autumn (September to November), and winter (December to February) were used to categorize the seasons. The district was divided into eastern, central, and western regions, while the region was divided into urban and rural sections [8]. Anemia is judged by the Chinese standard for anemia screening [9]. The Chinese adult criterion for abdominal obesity was used to categorize the BMI level as thin, normal, overweight, and obese [10]. Primary school and below, junior high school, high school, secondary school, and junior college or above are the different categories for education level. The two categories of marital status are married and single, divorced, or widowed.

### 2.3. Sample Detection

The fasting venous blood was drawn and centrifuged (3000 rpm, 15 min) for 30 min after blood taken. The serum was separated, divided, and stored in a brown vessel at a temperature of –20 °C in the lab where the survey was conducted. Every blood sample from every survey location was sent by cold chain to the bio specimen bank at NINH, China CDC. All of the samples were kept in a −70 °C freezer prior to sample detection. We measured the serum concentration of 25(OH)D for selected samples by ELISA kits (Immuno Diagnostic System Ltd., Boldon, UK). Quality control was performed by using the kit’s own quality control products and 10% parallel samples. The coefficient of variation (CV) of the parallel samples was 4.77%. The CV of the quality control product in the kit was 6.66% and 6.92% at high and low values, respectively, corresponding to a deviation of 1.23% and 2.73%.

Serum 25(OH)D values of 20 ng/mL or more are regarded as adequate. Serum 25(OH)D concentrations below 12 ng/mL were defined as deficient (VDD), and those more than 12 ng/mL but below 20 ng/mL were defined as inadequate (VDI) [11]. VDD and VDI were defined as vitamin D inadequacy (less than 20 ng/mL) in this study, whereas 25(OH)D concentrations more than or equal to 20 ng/mL were termed vitamin D sufficiency.

### 2.4. Data Analyses

All participants throughout this study were separated into several sub-groups based on various postulated influencing variables. The concentration of 25(OH)D was recorded using P50(P25–P75), due to inconsistency with the normal distribution. The Kruskal–Wallis test was used to compare the differences. Before calculating the prevalence, we combined the sampling design of the samples with demographic information from China’s Sixth Population Census to determine the final individual weights. The prevalence of VDD, VDI, and vitamin D sufficiency were expressed as percentages (%), and prevalence between subgroups were compared using the chi square test. The relationship between vitamin D deficiency and potential influencing factors (e.g., age, gender, ethnicity, district, region type, latitude, season, BMI, abdominal obesity, anemia, education, marital status) was examined using multivariable logistic regression analysis, and the odds ratio (OR) and 95% confidence intervals (CIs) were calculated. Considering the high deficiency and insufficiency of vitamin D, we also calculated the prevalence ratio (PR) and the 95% CIs by log-binomial models [12]. All of the associated variables were then adjusted to analyze changes in vitamin D quartiles. SAS 9.4 software (SAS Institute, Cary, NC, USA) was used to analyze all of the data. With *p* < 0.05, the difference was statistically significant.

## 3. Results

### 3.1. Basic Characteristics

In total, 6273 Chinese older adults (3130 males and 3143 females) were included in this study (Table 1). They were 73.3 years old (interquartile range (IQR) 64.8–77.4 y, range 60.0–97.0 y). The age groups of 60–69, 70–79, and 80 years old and above accounted for 43.46%, 42.52%, and 14.03%, respectively. Overall, 38.79% (2433) participants were from urban areas and 61.21% (3810) were from rural areas. Participants from the midlands, the east, and the west accounted for 33.41% (2096), 30.34% (1903), and 36.25% (2274), respectively. The median BMI was 23.33 kg/m^2^ (IQR 21.01–25.88); 42.26% older adults were overweight or obese, and more than 50% older adults were pre-abdominally obese or abdominally obese. Overall, 10.30% (646) participants were anemic. Only 2.12% of blood samples were taken in summer and 11.80% in spring, while 43.65% were in winter and 42.44% were in autumn. A total of 7.33% (460) participants were from tropical areas, 40.76% (2557) were from subtropical areas, 36.94% (2317) were from warm temperate, and 14.97% (939) were from medium temperate areas. In total, 81.97% (8142) of the participants were married.

### 3.2. Nutritional Status of Vitamin D

The median blood 25(OH)D concentration for Chinese older persons is 18.48 ng/mL (IQR 13.27–24.71 ng/mL) (Table 1). Males exhibited 25(OH)D concentrations that were significantly greater than those of females. With aging, the median serum 25(OH)D level considerably dropped. The concentration of 25(OH)D in Han ethnicity was significantly higher than that in minority ethnicities. Urban participants had significantly lower median 25(OH)D concentrations than rural participants. Participants from the eastern region had the greatest median level of 25(OH)D, followed by those from the middle region, and those from the western region had the lowest. The highest median 25(OH)D concentrations were found in the older adults in the tropics, followed by the subtropics and medium and warm temperate zones. Seasonally, summertime had the greatest median 25(OH)D concentrations while wintertime observed the lowest. As BMI increases, 25(OH)D concentrations in the participants decrease, with the lowest concentration being in obese individuals. Similarly, the higher the degree of abdominal obesity, the lower the 25(OH)D concentration. Older adults with higher levels of education had lower 25(OH)D concentrations. The 25(OH)D concentration of the older adults with college education and above was significantly lower than those with primary education. Married individuals had higher 25(OH)D concentrations than those who were unmarried or living alone.

Chinese older persons’ prevalence of vitamin D deficiency, insufficiency, and sufficiency in CACDNS 2015–2017 was 22.17%, 36.41%, and 41.73%, respectively (Table 1). Higher prevalence of vitamin D sufficiency was significant in males (50.65%) rather than in females (33.50%, *p* < 0.001). Older adults aged 60–69.9 years have much lower VDD prevalence than the other age groups. The VDD prevalence of minority ethnicity was higher than that of Han ethnicity. The prevalence of vitamin D sufficiency of the older adults in urban areas is significantly lower than those in rural areas. The VDD prevalence increased significantly from the eastern region to the midlands and western regions. The lowest VDD prevalence was found in the tropical areas, then followed by subtropical, medium temperate, and warm temperate areas. Obese older adults had the highest VDD prevalence, followed by thin and overweight people, while those with normal BMI had the lowest VDD prevalence. In terms of vitamin D sufficiency, as BMI increases, the rate of vitamin D sufficiency decreases. With the increase in degree of abdominal obesity, vitamin D sufficiency rate decreased. VDD prevalence was highest in winter and lowest in summer. Married people had lower VDD prevalence than those not married or living alone.

### 3.3. Influencing Factors for Vitamin D Inadequacy

Inadequate vitamin D status (25(OH)D < 20 ng/mL) was associated with factors including female, 80 years and older, ethnic minorities, living in urban, western areas, subtropical zone, warm temperate zone, medium temperate zone, abdominally obese, and anemia based on the results of PR and OR (Table 2). In addition to the above factors, the results of PR showed that being 70–79 years old, living in midlands, and having a middle or college education level also increased the probability of having vitamin D inadequacy.

Latitude has the greatest impact on vitamin D inadequacy. Among the factors mentioned above, residence in the warm temperate zone had the highest occurrence of vitamin D inadequacy compared to the tropical zone, which was the influencing factor with the highest OR value and PR value, then followed by the medium temperate zone and subtropical zone, compared to tropical zone).

Besides these fixed factors, we found that abdominal obesity and anemia increased the probability of vitamin D inadequacy. The influencing factors were consistent for both sexes, according to the findings of the logistic regressions model and log-binomial model carried on the factors mentioned above.

## 4. Discussion

In this study, we found that the median concentration of 25OH)D in Chinese older adults was lower than 5 years ago [13]. The prevalence of VDD in Chinese older adults was 22.17%, higher than the total rate of Chinese adult aged 18 years and above, 21.4% [14], and it was also significantly higher than 5 years ago (VDD prevalence was 9.7% in 2010–2013, unpublished data). The probability of vitamin D deficiency and insufficiency is increased in women, people over the age of 70, ethnic minorities, living in urban areas, the midlands, or western areas, warm or medium temperate zones, those with abdominal obesity or anemia, and those who have completed middle school or higher education.

Consistent with all other age groups [14,15], older women also had significantly higher rates of vitamin D deficiency and insufficiency than that of men. VDD was significantly higher in those over 70 years old than in those under 70 years old. The difference in age may be due to the aging and chronic diseases that make the older adults less active, thus leading to insufficient vitamin D synthesis. The tendency for 25(OH)D levels to decline with age was also found in Croatian postmenopausal women [16]. The older adults living in rural areas had a significantly lower probability of VDD than those living in urban areas. This is consistent with what has been seen previously in other age groups [14]. Air pollution, which is more severe in urban areas, reduces ambient UV levels, which affects vitamin D levels in the population. The prevalence of VDD from the east to the west shows a gradual increase, but this does not imply its variation with longitude, as the division between east, central, and west is more based on economic level. The VDD rate is much higher among ethnic minorities, which may be related to ethnic culture and dress requirement aspects. The effect of latitude and season on VDD is highly significant, as it is directly related to sunshine intensity. The VDD prevalence in the tropics is only 2.4%, much lower than in the other temperate zones. As we all know, VDD rates are lowest in the summer months because of sufficient sunlight; the VDD rate in summer was less than half of that in spring and winter in the older adults of this study. We also found that the rate of VDD in the married or partnered population was significantly lower than that of the single, unmarried population, probably because the married population was more regular in diet, outdoor activities, etc., owing to the companionship and supervision of their partners.

In addition to these fixed factors such as region, age, season, ethnicity, and latitude that affect 25(OH)D levels, we also found abdominal obesity and anemia might increase the probability of vitamin D inadequacy by 21% and 38%, respectively. However, no association between BMI and vitamin D inadequacy was found in our study. Evidence from published meta-analyses and several studies agrees that the inverse relationship between vitamin D and obesity is clear [17,18], and several studies have attempted to explain or shed light on the etiology and effects of vitamin D and obesity, but no consistent conclusions have been reached and there are no in-depth studies on abdominal obesity and vitamin D. As for anemia, Zhang et al. reported that low 25(OH)D levels were associated with increased risk of anemia [19]. In short, there is no accepted view on the correlation between either abdominal obesity and vitamin D or anemia and vitamin D, so more in-depth studies are still needed to shed light on this in the future.

There are some limitations to this study. We focused more on the nutritional status of vitamin D within the older adults and did not address its sources including each participant’s level of sunlight exposure, or dietary sources of vitamin D. In addition, physical activities, especially the duration and intensity of outdoor activities, were not included because of poor quality of data. The factors mentioned above will lead to bias. Nevertheless, our study has some clear benefits. We presented extensive, nationally representative data on older adults’ vitamin D nutritional status using scientific sampling techniques and stringent quality control procedures. Since the rate of vitamin D inadequacy in Chinese older adult was higher than 10% [20], we also calculated PRs to analyze the factors and compared them with ORs, and found that PRs were indeed more intuitive and identified the variables more comprehensively.

## 5. Conclusions

In conclusion, vitamin D deficiency and insufficiency were quite common among older adults in China, accounting for 58.27%, which is nearly 20% worse compared to that of five years ago. Women, people aged above 70 years, ethnic minorities, people living in urban areas, midlands, or western areas, warm or medium temperate zones, with middle school and above education level, and people with abdominal obesity and anemia have an increased chance of vitamin D deficiency and insufficiency. We encourage Chinese older adults to engage in more outdoor activities to obtain enough sunshine, especially for those with increased prevalence probability. Increased intake of dietary vitamin D and supplements, consistent with the Chinese Dietary Reference Intakes, is also encouraged.

## Figures and Tables

**Table 1 nutrients-15-04476-t001:** Vitamin D nutritional status for older adults from the China Adult Chronic Disease and Nutrition Survey 2015–2017.

Characteristics	No.	25(OH)D Concentration (ng/mL)		Vitamin D Nutritional Status (%, 95%CI)			
		Median (P25–P75)	*p* Value	Sufficiency	Insufficiency	Deficiency	** *p* ** **Value**
Total	6273	18.48 (13.27–24.71)		41.73 (38.52–44.93)	36.10 (34.20–38.00)	22.17 (19.58–24.76)	
Sex			<0.0001				<0.0001
Male	3130	20.30 (14.72–26.54)		50.65 (47.06–54.24)	33.18 (30.75–35.60)	16.17 (13.75–18.59)	
Female	3143	16.86 (12.08–22.56)		33.50 (30.14–36.87)	38.79 (36.43–41.16)	27.70 (24.60–30.80)	
Age group			<0.0001				<0.0001
60–69 years	2726	18.96 (13.88–25.11) ^a^		44.08 (40.62–47.53)	36.14 (33.85–38.44)	19.78 (17.08–22.47)	
70–79 years	2667	18.15 (12.82–24.28)		38.69 (34.96–42.43)	35.95 (33.46–38.44)	25.35 (22.11–28.59)	
80 years+	880	17.65 (12.87–24.57)		38.93 (34.69–43.17)	36.41 (32.98–39.84)	24.66 (21.14–28.19)	
Ethnicity			<0.0001				0.0018
Han	5565	18.67 (13.58–24.91)		42.80 (39.49–46.11)	36.48 (34.50–38.46)	20.72 (18.17–23.27)	
Minorities	708	16.73 (11.52–22.94)		33.23 (24.05–42.42)	33.09 (27.43–38.75)	33.68 (24.78–42.58)	
Region type			<0.0001				0.0004
Urban	2433	17.19 (12.62–23.09)		36.93 (33.07–40.80)	38.63 (35.97–41.29)	24.44 (20.99–27.88)	
Rural	3840	19.32 (13.77–25.79)		45.52 (41.51–49.54)	34.10 (31.85–36.35)	20.38 (17.26–23.49)	
District			<0.0001				<0.0001
Eastern	2096	20.77 (15.29–27.12) ^a^		50.81 (45.02–56.61)	35.21 (31.38–39.05)	13.98 (10.76–17.19)	
Midlands	1903	18.78 (13.65–24.86) ^b^		43.34 (37.98–48.70)	36.35 (33.25–39.46)	20.31 (15.94–24.68)	
Western	2274	16.25 (11.33–22.14) ^c^		31.76 (26.97–36.56)	36.73 (33.90–39.57)	31.50 (26.62–36.38)	
Latitude			<0.0001				<0.0001
Tropical	460	27.57 (23.09–33.71) ^a^		83.79 (75.47–92.10)	13.82 (6.26–21.37)	2.40 (0.59–4.21)	
Subtropical	2557	21.19 (16.14–26.84) ^b^		55.04 (50.42–59.66)	33.40 (30.43–36.37)	11.56 (8.68–14.43)	
Warm temperate	2317	15.05 (10.84–20.13) ^d^		24.34 (20.98–27.71)	41.13 (38.55–43.71)	34.53 (30.15–38.91)	
Medium temperate	939	16.22 (11.97–21.23) ^c^		30.25 (25.51–34.99)	41.10 (37.30–44.90)	28.65 (22.86–34.44)	
BMI			<0.0001				<0.0001
Thin	396	20.01 (13.35–25.77)		47.77 (40.97–54.56)	28.90 (23.66–34.13)	23.34 (18.29–28.39)	
Normal	3226	19.15 (13.73–25.88)		45.34 (41.54–49.14)	33.82 (31.54–36.11)	20.84 (17.95–23.73)	
Overweight	1905	17.95 (13.27–23.40)		38.45 (35.04–41.86)	39.64 (37.04–42.25)	21.91 (18.97–24.84)	
Obese	746	17.05 (12.07–22.33)		33.37 (29.07–37.66)	39.18 (35.11–43.24)	27.46 (22.75–32.16)	
Abdominal obesity			<0.0001				<0.0001
No	3089	19.55 (14.06–26.08) ^a^		47.10 (43.26–50.93)	33.71 (31.27–36.14)	19.20 (16.50–21.89)	
Pre-abdominal obesity	1109	18.55 (13.10–24.55) ^b^		42.23 (38.03–46.44)	35.28 (31.74–38.83)	22.48 (18.87–26.09)	
Abdominal obesity	2075	17.26 (12.39–22.56) ^c^		34.08 (30.83–37.33)	39.82 (37.30–42.34)	26.10 (22.69–29.51)	
Anemia			0.307				0.456
No	5627	18.54 (13.30–24.75)		41.97 (38.74–45.20)	36.12 (34.16–38.09)	21.91 (19.27–24.55)	
Yes	646	18.00 (13.04–24.37)		39.40 (32.99–45.81)	35.90 (31.70–40.11)	24.70 (19.34–30.06)	
Season *			<0.0001				<0.0001
Spring	740	17.59 (12.65–23.90) ^b^		37.94 (28.59–47.29)	37.74 (31.42–44.07)	24.32 (16.97–31.66)	
Summer	133	20.58 (15.90–25.42) ^a^		53.29 (39.02–67.56)	35.55 (22.69–48.42)	11.15 (2.14–20.16)	
Autumn	2662	19.41 (14.39–25.17) ^a^		46.67 (42.70–50.65)	37.25 (34.82–39.69)	16.07 (13.43–18.71)	
Winter	2738	17.51 (12.10–24.23) ^b^		37.45 (32.96–41.94)	34.58 (31.94–37.22)	27.97 (23.85–32.10)	
Education			0.032				0.158
Primary	4754	18.62 (13.25–24.95) ^a^		42.13 (38.62–45.64)	35.02 (33.01–37.04)	22.85 (19.96–25.73)	
Middle	1355	18.27 (13.55–24.31) ^ab^		41.03 (37.04–45.01)	38.81 (35.64–41.98)	20.16 (16.85–23.47)	
College	164	17.30 (12.54–22.19) ^b^		37.20 (27.74–46.67)	40.97 (32.09–49.84)	21.83 (14.34–29.32)	
Marriage			<0.0001				<0.0001
Yes	5142	18.78 (13.60–24.93)		43.02 (39.73–46.31)	35.78 (33.74–37.82)	21.20 (18.63–23.78)	
No	1131	17.01 (12.09–23.34)		34.65 (30.46–38.84)	37.86 (34.52–41.21)	27.48 (23.52–31.45)	

* The months of March, April, and May are classified as spring; June, July, and August as summer; September, October, and November as autumn; and December, January, and February as winter. Substantial differences between groups are denoted by the letters ^a^, ^b^, ^c^, and ^d^; different letters denote a substantial difference.

**Table 2 nutrients-15-04476-t002:** Odds ratio and prevalence ratio of influencing factors for Vitamin D inadequacy in CACDNS 2015–2017.

Characteristics	Odds Ratio (OR)		Prevalence Ratio (PR)	
	OR (95%CI)	*p* Value	PR (95%CI)	*p* Value
Sex				
Male	Ref		Ref	
Female	2.43 (2.10–2.80)	<0.0001	2.46 (2.18–2.79)	<0.0001
Age group				
60–69 years	Ref		Ref	
70–79 years	1.31 (1.14–1.50)	0.295	1.30 (1.14–1.47)	<0.0001
80 years+	1.48 (1.22–1.81)	0.005	1.49 (1.24–1.80)	<0.0001
Ethnicity				
Han	Ref		Ref	
Minorities	1.61 (1.10–2.36)	0.014	1.58 (1.29–1.94)	<0.0001
Region type				
Rural	Ref		Ref	
Urban	1.46 (1.21–1.77)	0.0001	1.49 (1.31–1.69)	<0.0001
District				
Eastern	Ref		Ref	
Midlands	1.34 (1.01–1.76)	0.069	1.34 (1.16–1.55)	<0.0001
Western	2.82 (2.11–3.75)	<0.0001	3.03 (2.61–3.53)	<0.0001
Latitude				
tropical	Ref		Ref	
Subtropical	4.82 (2.42–9.59)	0.010	5.16 (3.87–6.96)	<0.0001
Warm temperate	23.08 (11.57–46.03)	<0.0001	24.34 (18.17–33.02)	<0.0001
Medium temperate	16.55 (7.96–34.39)	<0.0001	18.26 (13.27–25.40)	<0.0001
BMI				
Thin	1.16 (0.88–1.53)	0.292	1.12 (0.88–1.44)	0.354
Normal	Ref		Ref	
Overweight	1.02 (0.87–1.20)	0.944	1.01 (0.87–1.18)	0.883
Obese	0.93 (0.72–1.21)	0.341	0.94 (0.74–1.18)	0.574
Abdominal obesity				
No	Ref		Ref	
Pre-abdominal obesity	1.09 (0.91–1.30)	0.746	1.09 (0.92–1.30)	0.303
Abdominal obesity	1.24 (1.03–1.50)	0.038	1.21 (1.02–1.45)	0.032
Anemia				
No	Ref		Ref	
Yes	1.37 (1.10–1.70)	0.005	1.38 (1.14–1.67)	0.001
Season				
Spring	1.07 (0.55–2.10)	0.330	1.09 (0.70–1.69)	0.700
Summer	Ref			
Autumn	0.70 (0.36–1.37)	0.010	0.72 (0.48–1.08)	0.115
Winter	1.05 (0.53–2.05)	0.353	1.07 (0.71–1.61)	0.759
Education				
Primary	Ref		Ref	
Middle	1.21 (1.00–1.45)	0.717	1.24 (1.06–1.44)	0.006
College	1.33 (0.86–2.04)	0.381	1.47 (1.01–2.16)	0.047
Marriage				
Yes	Ref		Ref	
No	1.16 (0.97–1.39)	0.096	1.17 (1.00–1.38)	0.054

## Data Availability

All data for this article are from CACDNS 2015–2017 and specific data are not publicly available at this time.

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
