# Peer review of "Vitamin D Status and Associated Factors of Older Adults in the Cross-Sectional 2015–2017 Survey"

_nutrients, 2023, doi:10.3390/nu15204476_

Round 1
Reviewer 1 Report
The topic addressed in this manuscript notably interests nutritional epidemiology and public health. Certain significant concerns must be considered for enhancing the quality of the manuscript, particularly the statistical analysis.
a) The authors ought to include information on the main findings with values in the abstract.
b) The introduction should elaborate on the previous literature evidence about the connection between vitamin D and obesity. Please, rewrite this section.
c) Was information collected on physical activity and diet? This information would be necessary as confounders to include in your models.
d) In this cross-sectional study would be better performance prevalence ratio instead of odds ratio.
Please, the authors can see the following references:
Espelt A., Marí-Dell’Olmo M., Penelo E., Bosque-Prous M. Applied Prevalence Ratio estimation with different Regression models: An example from a cross-national study on substance use research. Adicciones. 2016; 29:105–112. doi: 10.20882/adicciones.823.
Barros A.J.D., Hirakata V.N. Alternatives for logistic regression in cross-sectional studies: An empirical comparison of models that directly estimate the prevalence ratio. BMC Med. Res. Methodol. 2003; 3:21. doi: 10.1186/1471-2288-3-21
e) When using a sampling method through a multi-stage, stratified, cluster-random sampling procedure, it is important to consider weighting in regression analysis, especially if you intend to generalize your results to the broader population based on the selected sample. Kindly incorporate information related to this issue in the statistical analysis.
f) The first paragraph of the discussion should include a summary of the study’s main findings.
1. The discussion section of the article would be improved by conducting a comprehensive comparison between the results of your study and prior research, detailing potential differences and similarities. Additionally, it is crucial to link the discussion with the results. For example, the initial paragraph of the discussion is not connected to the results.
h) The information included in the limitation of study is imprecise and it should consist of information considering sources of potential bias or imprecision. Discuss both the direction and magnitude of any potential bias. In addition, they should include information on the strength of the study.
i) The conclusion did not respond to the research question. Please, rewrite this section in this sense.
Author Response
Point 1:The authors ought to include information on the main findings with values in the abstract.
Response to Point 1: We have revised the content and the title, and the revised manuscript mainly demonstrates the vitamin D nutritional status and the risk factors of vitamin D insufficiency and deficiency in Chinese older adults. Regarding the results and discussion section on vitamin D and obesity, after careful consideration, we believe that it is difficult to reveal the causal relationship in such a cross-sectional study, and more in-depth studies need to be conducted to reveal it, which is one of the directions of our future research endeavors. Therefore, in this revised manuscript, we only show the effect of obesity on vitamin D inadequacy through PR/OR values. In the abstract part, we have added results about PR values accordingly.
Point 2:The introduction should elaborate on the previous literature evidence about the connection between vitamin D and obesity. Please, rewrite this section.
Response to Point 2: The priority purpose of this manuscript was to report vitamin D levels in Chinese older adults. We also analyzed the related impact factors for vitamin D deficiency and insufficiency, during the analysis, we found a significant correlation between abdominal obesity and vitamin D levels. We had intended to demonstrate the link between the two, but after your kind reminder, we recognized that the information in this study does not make our study objective apparent enough, so we deleted part 3.4 Association between vitamin D level and obesity/abdominal obesity in the revised manuscript and made revision in the discussion section. Therefore, we did not elaborate on the previous literature evidence about the connection between vitamin D and obesity in the introduction part in the revised version.
Point 3:Was information collected on physical activity and diet? This information would be necessary as confounders to include in your models.
Response to Point 3: Information about physical activity and diet were included in the CACDNS 2015-2017, mainly based on self-report. The data itself is biased and, in the case of physical activity, does not differentiate between the duration and intensity of outdoor and indoor activities, making it difficult to include in the analysis of this study. We have added these factors to the limitations section.
Point 4:In this cross-sectional study would be better performance prevalence ratio instead of odds ratio.
Please, the authors can see the following references:
Espelt A., Marí-Dell’Olmo M., Penelo E., Bosque-Prous M. Applied Prevalence Ratio estimation with different Regression models: An example from a cross-national study on substance use research. Adicciones. 2016; 29:105–112. doi: 10.20882/adicciones.823.
Barros A.J.D., Hirakata V.N. Alternatives for logistic regression in cross-sectional studies: An empirical comparison of models that directly estimate the prevalence ratio. BMC Med. Res. Methodol. 2003; 3:21. doi: 10.1186/1471-2288-3-21
Response to Point 4: Thank you so much for your helpful advice. Based on your recommendation, we read the two papers and reviewed the related information and learned that OR can represent PR well when the prevalence is less than 10%, but when the prevalence is higher the OR value will overestimate the PR value. In our study, the results show a great degree of agreement between OR and PR in terms of factors that increase the prevalence of vitamin D inadequacy, but there are also differences, such as level of education. Overall, the confidence intervals for PR are more centered and the results are easier to interpret directly. However, considering that OR is still mostly expressed in literatures on the similar topic, we retained PR and OR in the revised manuscript.
Point 5:When using a sampling method through a multi-stage, stratified, cluster-random sampling procedure, it is important to consider weighting in regression analysis, especially if you intend to generalize your results to the broader population based on the selected sample. Kindly incorporate information related to this issue in the statistical analysis.
Response to Point 5: Thank you for your reminder. We weighted the data when analyzing it, but did not describe it in the section on statistical analysis, which we have added in the revised version.
Point 6: The first paragraph of the discussion should include a summary of the study’s main findings.
The discussion section of the article would be improved by conducting a comprehensive comparison between the results of your study and prior research, detailing potential differences and similarities. Additionally, it is crucial to link the discussion with the results. For example, the initial paragraph of the discussion is not connected to the results.
Response to Point 6: Thank you for your suggestion. We have revised the discussion section.
In the first paragraph, we displayed the main results of this study, the levels and prevalence of vitamin D and its comparison with the previous round of survey.
The 2nd and 3rd paragraph analyzes the factors influencing vitamin D deficiency and insufficiency. There are few national-level reports of vitamin D nutritional status in older adults in the literature, and we tried to compare our results with what has been reported.
Point 7: The information included in the limitation of study is imprecise and it should consist of information considering sources of potential bias or imprecision. Discuss both the direction and magnitude of any potential bias. In addition, they should include information on the strength of the study.
Response to Point 7: Thank you for your suggestion. We have elaborated on the limitations of this study, such as physical activity, diet, etc., as you mentioned. We also emphasize the strengths of this study.
Point 8:The conclusion did not respond to the research question. Please, rewrite this section in this sense.
Response to Point 8: Thank you for your suggestion. We have revised the conclusion part.
Reviewer 2 Report
The article is devoted to the search and study of associations of blood levels of vitamin D with obesity in elderly Chinese residents.
My comments:
1. The relevance of the topic is questionable. What is the purpose of studying vitamin D levels in older people?
2. In the Abstract, the authors indicate that certain factors, including obesity and anemia, significantly increase the risk of vitamin D deficiency. The risk was not studied in the article. The study was not prospective. Only connections and associations were studied.
3. The practical significance of the study is unclear. The authors conclude that older people need to get out in the sun, be physically active and fight obesity. And what? These are well-known recommendations.
I have doubts about the article, its relevance and practical significance. On the other hand, the authors can rework the article and explain these issues in detail. Then the article will be more understandable.
Author Response
Point 1. The relevance of the topic is questionable. What is the purpose of studying vitamin D levels in older people?
Response to Point 1: The priority purpose of this manuscript was to report vitamin D levels in Chinese older adults. We also analyzed the related impact factors for vitamin D deficiency and insufficiency, during the analysis, we found a significant correlation between abdominal obesity and vitamin D levels. We had intended to demonstrate the link between the two, but after your kind reminder, we recognized that the information in this study does not make our study objective apparent enough, so we deleted part 3.4 Association between vitamin D level and obesity/abdominal obesity in the revised manuscript and made revision in the discussion section.
Point 2. In the Abstract, the authors indicate that certain factors, including obesity and anemia, significantly increase the risk of vitamin D deficiency. The risk was not studied in the article. The study was not prospective. Only connections and associations were studied.
Response to Point 2: It is true, as you pointed out, that we only found abdominal obesity and anemia to be risk factors for vitamin D inadequacy by logistics regression, and since this is a cross-sectional study, this study only found such a connection, but it really did not study this risk in more depth, and therefore, we have revised the content as we answered in response to point 1 referring to obesity and anemia. And we also changed the wording throughout the text.
Point 3. The practical significance of the study is unclear. The authors conclude that older people need to get out in the sun, be physically active and fight obesity. And what? These are well-known recommendations.
Response to Point 3: Thank you for your kind suggestion, what we preferred to show in this study is the current status of vitamin D nutritional status and the severity of vitamin D insufficiency and deficiency, and thus the suggestion is also a reminder in conjunction with the risk factors found in this study and common sense.
Point 4. I have doubts about the article, its relevance and practical significance. On the other hand, the authors can rework the article and explain these issues in detail. Then the article will be more understandable.
Response to Point 4: Thank you very much for your kind reminder. We have revised the paper to show the vitamin D nutritional status of older adults in China, which we would want to emphasize. It is hoped that the findings of the present study would be beneficial in developing policy recommendations for vitamin D deficiency.
Round 2
Reviewer 1 Report
Thank you very much for your responses to all the points proposed. Below we include some minor changes that you should make.
a) Include the associated factors in the title that was studied as well.
b) Include the design of the study in the title.
c) Include the aim of the study in the abstract and in the last paragraph of the introduction.
d) Add to the abstract the main results of the associated factors.
e) It is unnecessary to present the results for OR and RP. From my point of view, it would be better to just present RP.
f) In the first paragraph of the discussion, include the main results about associated factors.
Author Response
Point 1: Include the associated factors in the title that was studied as well.
Response to Point 1: Thank you for your advice. We have revised the title into “Vitamin D Status and Associated Factors of Chinese Older Adults in the Cross-Sectional 2015-2017 Survey”.
Point 2: Include the design of the study in the title.
Response to Point 2: Thank you for your advice. We have revised the title into “Vitamin D Status and Associated Factors of Chinese Older Adults in the Cross-Sectional 2015-2017 Survey”.
Point 3: Include the aim of the study in the abstract and in the last paragraph of the introduction.
Response to Point 3: We have revised the abstract part and the last paragraph of the introduction.
Point 4: Add to the abstract the main results of the associated factors.
Response to Point 4: In the abstract section, limited to space requirements, we summarize our findings on the associated factors in the abstract section, presenting "The likelihood of vitamin D deficiency and insufficiency is increased in women, people aged and above 70 years, ethnic minorities, people living in urban areas, midlands or western areas, warm or medium temperate zones, with middle school and above education level, and people with abdominal obesity and anemia would increase the possibility of vitamin D deficiency and insufficiency with latitude having the greatest impact on vitamin D deficiency and insufficiency."
Point 5: It is unnecessary to present the results for OR and RP. From my point of view, it would be better to just present RP.
Response to Point 5: With your kindness suggestion, we followed the research and discussions on OR and PR and learned that there is still no consensus on the discussion of the applicability of PR and OR in cross-sectional studies.
As suggested in the two papers you recommended, PR is more intuitive than OR, which is a good estimate of PR when prevalence is low, but OR overestimates PR when prevalence is high. However, we also note that an alternative viewpoint has been expressed in other papers [1], when the assumptions for causal inference are met in cross-sectional studies, considering that neither PR nor OR can be used to correctly estimate cumulative incidence ratio (CIR) and that OR is the only metric that provides an unbiased estimate of the cumulative incidence ratio (CIR), it is considered that OR obtained using logistic regression modeling is still an appropriate choice for capturing incidence between exposure groups.
Based on the above, we considered whether both estimates could still be retained in the manuscript.
[1] Reichenheim, M. E., & Coutinho, E. S. (2010). Measures and models for causal inference in cross-sectional studies: arguments for the appropriateness of the prevalence odds ratio and related logistic regression. BMC medical research methodology, 10, 66.
Point 6: In the first paragraph of the discussion, include the main results about associated factors.
Response to Point 6: We have revised added the main results about associated factors in the first paragraph of the discussion.
Reviewer 2 Report
The authors corrected the article according to my comments. I have no further comments.
Author Response
Thank you very much for your help.